# A Hybrid Model for Cardiac Perfusion: Coupling a Discrete Coronary Arterial Tree Model with a Continuous Porous-Media Flow Model of the Myocardium

**DOI:** 10.3390/e25081229

**Published:** 2023-08-18

**Authors:** João R. Alves, Lucas A. Berg, Evandro D. Gaio, Bernardo M. Rocha, Rafael A. B. de Queiroz, Rodrigo W. dos Santos

**Affiliations:** 1Department of Education, Federal Institute of Education, Science and Technology of Mato Grosso, Sorriso 78895-150, Brazil; 2Department of Computer Science, Federal Univesity of Juiz de Fora, Juiz de Fora 36036-900, Brazilevandrodgaio@gmail.com (E.D.G.); bernardomartinsrocha@ice.ufjf.br (B.M.R.); 3Department of Computer Science, University of Oxford, Oxford OX3 7LD, UK; 4Departament of Computing, Federal Univesity of Ouro Preto, Ouro Preto 35400-000, Brazil

**Keywords:** cardiac perfusion, fractional flow reserve, cardiac modeling, cardiovascular diseases

## Abstract

This paper presents a novel hybrid approach for the computational modeling of cardiac perfusion, combining a discrete model of the coronary arterial tree with a continuous porous-media flow model of the myocardium. The constructive constrained optimization (CCO) algorithm captures the detailed topology and geometry of the coronary arterial tree network, while Poiseuille’s law governs blood flow within this network. Contrast agent dynamics, crucial for cardiac MRI perfusion assessment, are modeled using reaction–advection–diffusion equations within the porous-media framework. The model incorporates fibrosis–contrast agent interactions and considers contrast agent recirculation to simulate myocardial infarction and Gadolinium-based late-enhancement MRI findings. Numerical experiments simulate various scenarios, including normal perfusion, endocardial ischemia resulting from stenosis, and myocardial infarction. The results demonstrate the model’s efficacy in establishing the relationship between blood flow and stenosis in the coronary arterial tree and contrast agent dynamics and perfusion in the myocardial tissue. The hybrid model enables the integration of information from two different exams: computational fractional flow reserve (cFFR) measurements of the heart coronaries obtained from CT scans and heart perfusion and anatomy derived from MRI scans. The cFFR data can be integrated with the discrete arterial tree, while cardiac perfusion MRI data can be incorporated into the continuum part of the model. This integration enhances clinical understanding and treatment strategies for managing cardiovascular disease.

## 1. Introduction

Atherosclerotic cardiovascular diseases, the leading cause of death and disability worldwide, are strongly linked to various risk factors such as high blood pressure, smoking, diabetes, obesity, sedentary lifestyle, and high cholesterol levels. These conditions can lead to coronary atherosclerosis, aortic valve regurgitation, and left ventricle hypertrophy, resulting in reduced myocardium perfusion (MF), causing tissue damage (ischemia), and potentially leading to infarction. These diseases disproportionately affect individuals in low- and middle-income countries [1,2].

Assessing the cardiac circumstances of patients is critical for effective diagnosis and treatment. Medical imaging tools such as magnetic resonance imaging (MRI) and computed tomography (CT) play an important role in directly measuring blood perfusion, which is essential for understanding the patient’s cardiac situation.

In both MRI and CT, a contrasting agent (CA) is often administered to the patient. The resulting images generated by the chosen protocol highlight the poorly perfused regions of the heart, providing valuable information on the extent of ischemia and injury.

For cardiac tissue characterization in heart MRI, late gadolinium enhancement (LGE) is the most commonly used protocol. This technique enables the evaluation of myocardial scar formation and regional myocardial fibrosis by allowing gadolinium, the CA, to be perfused for approximately 600 s and adsorbed in areas of excess extracellular matrix. By highlighting areas of myocardial scarring and fibrosis, LGE can help identify patients at risk of arrhythmias or heart failure and guide treatment decisions.

Blood–tissue exchange investigation is a topic of long-standing interest to physiologists, and was first modeled mathematically as a set of reaction–diffusion–advection equations by [3]. More recently, ref. [4] proposed a framework for the simulation of cardiac perfusion using Darcy’s law using the idea of multicompartments to represent the different blood vessels’ spatial scale. In medical image analysis, ref. [5] proposed quantifying the behavior of contrast agents in MR perfusion imaging. The work used a simplified model of contrast agent transport and provided interesting insights on the design and selection of the appropriate CA for specific imaging protocols and postprocessing methods. Finally, ref. [6,7] proposed similar tools that also use mathematical models based on PDEs (partial differential equations) and images, in this case, from contrast-enhanced MRI exams. In particular, the previous work presented in [7] evaluated the CA dynamics for three different scenarios: healthy, ischemic, and infarct in a 2D mesh slice from an MRI exam, and performed a comparison with experimental data for the LGE protocol.

Fractional flow reserve (FFR) is a powerful technique for assessing cardiac perfusion. It measures the significance of narrowings in the coronary arteries that supply blood to the heart muscle, typically during an invasive coronary angiogram [8]. By assessing the blood flow across a specific area of narrowing, FFR can help determine whether the narrowing limits blood flow to the heart and potentially causes ischemia in the tissue. In silico FFR is a computational technique that can predict FFR measurements using blood flow models in the coronary arteries [9]. By integrating information from medical imaging with computational fluid dynamics, in silico FFR simulations can provide valuable information on coronary perfusion without invasive procedures. These simulations allow for the simulation of the physiological conditions of the patient’s heart and predict the hemodynamic significance of coronary narrowings, thus identifying those patients who would benefit most from interventions to improve blood flow, such as angioplasty or stenting.

In addition to the studies mentioned above, the following review on cardiac perfusion encompasses some biomedical research topics explored in the literature. Zhou et al. (2022) [10] present enzyme-free and enzyme-resistant detection methods for complement component 5, offering a valuable diagnostic tool for acute myocardial infarction. Hao et al. (2022) [11] introduced a new approach for evaluating cardiac ischemia/reperfusion using serum metal ion-induced cross-linking of photoelectrochemical peptides and proteins. Xue et al. (2022) [12] explored the impact of cardiomyocyte-specific knockout of ADAM17 on diabetic cardiomyopathy, revealing potential therapeutic strategies. Tian et al. (2022) [13] investigated the contribution of gut microbiome dysbiosis to abdominal aortic aneurysm through neutrophil extracellular trap formation. These studies collectively contribute to advancements in diagnostic techniques, treatment strategies, and understanding of the underlying mechanisms of various medical conditions.

The main objective of this study is to propose an extended model that accurately describes the perfusion of contrast agents in cardiac tissue by coupling a discrete coronary arterial network model with a porous-media flow model. We use Poiseuille’s law to mathematically model blood flow in the coronary arterial network to obtain intravascular pressure and velocity profiles. Intravascular dynamics of the contrast agent are described using reaction–diffusion–advection equations, while extravascular CA dynamics are modeled using reaction–diffusion equations. Our model considers the interaction of fibrosis (scar tissue) with the contrast agent to accurately simulate myocardial infarction. This interaction of CA and fibrosis is included in the model by adding an adsorption term to describe how CA is trapped in the fibrotic network. Our model also treats the tissue domain as an anisotropic porous media, taking into account the orientation of myocardial fibers, and includes the recirculation of the contrast agent in the body. A striking advantage of the proposed hybrid approach for cardiac perfusion is that it considers a discrete coronary arterial tree model, representing essential geometrical, topological, and fluid flow features for cardiac perfusion.

We perform numerical experiments to simulate scenarios of normal perfusion, endocardial ischemia due to stenosis, and myocardial infarction. Our results demonstrate that our model can effectively support noninvasive cardiac perfusion quantification via computer simulations. This study’s potential impact lies in its ability to integrate information from two different exams: fractional flow reserve (FFR) measurements of the heart coronaries from CT; and heart perfusion and topology from MRI scans [14,15].

## 2. Mathematical Modeling

Our previous paper [7] presented a cardiac perfusion model that exclusively relied on porous media. In that study, we comprehensively compared our simulations with clinical data, demonstrating a high degree of agreement between them.

In the current work, we adopted a distinct approach by incorporating a discrete arterial tree to capture intravascular dynamics alongside a continuous model for cardiac tissue. As a result, our primary objective in this study was to perform a qualitative validation of our methodology. To achieve this, we focused on examining the behavior of the contrast agent (CA) employed in cardiac MRI, specifically in response to various levels of stenosis present within the arterial tree.

### 2.1. The Hybrid Model

In previous work [7], the blood flow in the myocardium was considered as a single-phase flow in porous media. This approach can provide important information about the phenomenon. In that work, the Darcy system for flow in porous media was used to capture the blood pressure profiles and velocity in the intravascular tree that perfuses myocardium. In addition to that, a reaction–diffusion–advection equation was used, which describes the dynamics of the CA in scenarios of healthy and diseased tissue. Moreover, the paper proposes a third equation, a reaction–diffusion one, to describe the CA dynamics in two domains: intravascular and extravascular. This model can reproduce three important scenarios: perfusion in healthy tissue, an ischemic region, and an infarction. By the end, it reproduces the recirculation of the CA. Figure 1a summarizes those characteristics.

As said before, when it comes to an infarction, scars may build up with the rise of fibrosis to replace dead myocytes. In this way, when using both protocols of MRI contrast agent, two different situations can occur: In the first pass (FP) protocol, since it takes approximately 50 s, there is no time for the CA to reach the region of fibrosis. Thus, the images generated by the MRI indicate a dark color in the damaged region, whereas they indicate a bright color in the remaining regions. On the other hand, in the late-enhancement (LE) protocol, there is enough time for the CA to reach the area of fibrosis. Furthermore, the washing of CA is delayed, since the fibrosis network behaves like a trap for the CA. The phenomenon of fluid, particles, or substances sticking in a “solid phase” is called adsorption, and is depicted in the mathematical model in a third domain (fibrosis). This is detailed in Section 2.5. Figure 1b indicates a small region at the subendocardium that has been set up to simulate the scenario of infarction.

Nevertheless, the vessel network involves topological features that are anything but trivial. Thus, some information is not captured when a continuum porous medium represents the myocardium. Although such an approach can provide a useful understanding of myocardial perfusion, there are different conducts in modeling the phenomenon. This work proposes to replace the intravascular media, a continuum domain, with a discrete one. The constrained constructive optimization (CCO) [16,17] method was used to do so. This method allows one to generate models of peripheral vascular networks that are both detailed and realistic. Arterial tree models generated by CCO are able to mimic important properties of real arterial trees, such as segment radii [16,17], branching angle statistics [18] and pressure profiles [19,20].

### 2.2. Generating Optimized Arterial Tree Models

The optimized arterial tree models are generated by the CCO method [16,17], which is a well-established algorithm in the field of cardiovascular modeling that was used and validated in many works [21,22]. Below, we summarize the main features of the CCO method for ease of reference. It is based on the assumptions below:(A1)CCO trees grown under different optimization target functions show differences in structure that can be quantified by appropriately chosen numerical indexes [19]. In order to quantify the optimality of the CCO tree, the total intravascular volume is chosen for the optimization target function as recommended by other authors [16,17,23], according to the following equation:
(1)$$V=\pi \sum_{s=1}^{k_{tot}}l_{s}r_{s}^{2},$$
where $l_{s}$ and $r_{s}$ are the length and radius of the segment *s*, and $k_{tot}$ is the total number of segment in the tree. Segment means intervals of vessels between two consecutive bifurcations.(A2)The piece of tissue to be perfused is geometrically represented by a perfusion area or a perfusion volume.(A3)The arterial tree is modeled as a dichotomously branching system (binary tree) of straight cylindrical tubes representing the vessel segments, which are assumed to be rigid.(A4)The blood is modeled as an incompressible, homogeneous Newtonian fluid at steady state and laminar flow conditions.(A5)The flow resistance $R_{s}$ of each segment of the tree is assumed to follow Poiseuille’s law [24]:
(2)$$R_{s}=\left(\frac{8\eta }{\pi }\right)\frac{l_{s}}{r_{s}^{4}},$$
where η is the constant blood viscosity ($\eta =3.6\times 10^{-3}$ Pa s).(A6)The pressure drop $\Delta p_{s}$ along segment *s* is given by
(3)$$\Delta p_{s}=R_{s}Q_{s},$$
where $Q_{s}$ is the flow through segment *s*.

At each stage of development, a CCO tree model satisfies a set of physiological boundary conditions and constraints:(C1)Each terminal segment supplies an individual amount of blood flow $Q_{term}$ into the microcirculatory network, which is not modeled in detail.(C2)All terminal segments drain against a given, unique terminal pressure, $p_{term}$.(C3)The resistance of the resulting model tree induces a prespecified perfusion flow $Q_{perf}$ across the overall pressure gradient:
(4)$$\Delta p=p_{perf}-p_{term},$$
where $p_{perf}$ is the perfusion pressure at the inlet of the root segment (main feeding artery).(C4)At bifurcations, the radii of parent ($r_{0}$) and daughter segments ($r_{1}$, $r_{2}$) are forced to exactly fulfill a bifurcation law derived from real coronary trees [25]:
(5)$$r_{0}^{\gamma }=r_{1}^{\gamma }+r_{2}^{\gamma },$$
where γ is a constant exponent with ranging between 2.55 and 3, governing the shrinkage of radii across bifurcations. Minimum reflection of pulse waves at bifurcations is achieved with γ=2.55 [26]. For uniform shear stress, all over tree, γ should be set to 3 [27]. From Murray’s law [23], γ=3 is obtained as a necessary condition for minimum energy consumption in a vascular network.

### 2.3. Coupling Discrete and Continuous Models

In the hybrid model, the CCO method describes the volumetric flow $Q_{j}$, as well as the radius $r_{j}$ of every tree segment *j*. In other words, each segment *j* is a one-dimensional domain with its flow and radius values. The velocity $v_{j}$ of each segment *j* is given by:(6)$$v_{j}=\frac{Q_{j}}{\pi r_{j}^{2}}.$$

This velocity field of the arterial tree is used to calculate the dynamics of the intravascular concentration of the CA, $C_{i}$, with the following reaction–diffusion–advection equation:(7)$$\frac{\partial C_{i}^{j}}{\partial t}+\nabla \cdot v_{j}C_{i}^{j}-\nabla \cdot \left(D_{i}^{j}\nabla C_{i}^{j}\right)+f_{j}=0,$$
where $C_{i}^{j}$ is the intravascular contrast agent concentration at segment *j* of the arterial tree, and $D_{i}^{j}$ is the corresponding diffusion coefficient. For the simulation of the CA in the cardiac tissue, $C_{e}$, the extravascular domain is treated as a continuum via the theory of porous media. We present the model for the dynamics of the CA in the extravascular domain in the next section. The flux between intravascular and extravascular domains is given by the following term:(8)$$f_{j}=\left\{\begin{array}{c} P_{j}\left(C_{i}^{j}-C_{e}\right),\;if\;C_{i}^{j}>C_{e},\\ 0,\;\mathrm{otherwise}, \end{array}\right.$$
where $P_{j}$ is the endothelial permeability. In this hybrid model, the exchange of CA between intravascular and extravascular media takes place only at the terminal points of the tree. This exchange of CA occurs in a one-to-one way, between tissue points closest to the endpoints of the arterial tree; it is modeled by (Equation 8), and illustrated by Figure 2.

It is important to note that the CCO method generates the vascular structure down to the prearteriolar level. The exchange of oxygen and nutrients (perfusion) and the exchange of CA in an MRI exam, from the intravascular environment to the tissue, happens at the capillary level. In this way, the hybrid model is simplified to consider that the CA transition happens at the prearteriolar level. The results presented here are preliminary. However, they still allow for an essential analysis of the relationship between anatomical (stenosis) and functional (perfusion) problems.

### 2.4. Extravascular Model

There are many different ways to represent and deal with circulatory models in the literature [3,4,5]. In this work, the perfusion is modeled through a reaction–diffusion–advection equation in porous media.

In this work, we consider the cardiac tissue as a porous media, comprising an intravascular domain and an extravascular domain. The extravascular domain consists of two distinct components: the interstitial domain and, when applicable, a fibrotic domain. The intravascular region represents a combination of arteries and capillaries, whereas the extravascular region encompasses the interstitial space and, if present, a fibrotic region. The volume ratios of these different domains are taken as follows: (1) ϕ is the ratio between the volume of the intravascular domain (the arterial tree) and the total volume of the cardiac tissue; (2) λ represents the fraction of the extravascular domain that is occupied by the interstitial space; (3) and $\lambda_{f}$ is the fraction of the interstitium occupied by the region with fibrosis.

A system of diffusion–advection equations can describe contrast agent dynamics in both intra- and extravascular regions. The equations governing the dynamics of the concentrations of CA in the extravascular region, denoted $C_{e}$, are given by:(9)$$\begin{array}{ccc} & \frac{\partial (\left(1-\phi \right)\lambda C_{e})}{\partial t}-\left(1-\phi \right)\lambda \nabla \cdot \left(D_{e}\nabla C_{e}\right)-f+\left(1-\phi \right)\lambda k_{e}C_{e}+g=0, & \mathrm{in}\;\Omega_{e}, \end{array}$$
where $D_{i}$ and $D_{e}$ are diffusion tensors for the intra- and extravascular regions, respectively, *f* represents the communication between the domains, which is given by:(10)$$f=\left\{\begin{array}{cc} P(C_{i}-C_{e}),\; & if\;C_{i}>C_{e},\\ 0,\; & \mathrm{otherwise}, \end{array}\right.$$
where *P* is the endothelial permeability. The term $\left(1-\phi \right)\lambda k_{e}C_{e}$ models the flow from the interstitial space to the venous system, and λ represents the fraction of the extravascular domain that is occupied by the interstitial space.

### 2.5. Contrast Agent Adsorption

In addition, another equation and variable are needed to capture how the CA is trapped in the excess of extracellular matrix, which is the case of fibrosis or scar. This phenomenon, fluid (CA) attaching to a solid phase (extracellular matrix), is called adsorption, and is modeled by:(11)$$\frac{\partial (\left(1-\phi \right)\lambda \lambda_{f}C_{f})}{\partial t}+\left(1-\phi \right)\lambda \lambda_{f}k_{f}C_{f}-g=0,\;\mathrm{em}\;\Omega_{f},$$
where $C_{f}$ is the concentration of CA in the fibrosis network domain $\Omega_{f}$, *g* is a exchange term between the fibrotic and extravascular domains, and the $k_{f}C_{f}$ term describes the flow from the fibrotic network to the venous system. The exchange term *g* is given by:(12)$$g=\left(1-\phi \right)\lambda \lambda_{f}k_{ef}C_{e},$$
where $k_{ef}$ is the rate at which the contrast moves from the interstitium to the fibrosis, and $\lambda_{f}$ is the fraction of the interstitium occupied by the region with fibrosis.

### 2.6. Recirculation of the Contrast Agent

The cyclic behavior of the CA is captured during MRI or CT scans. Part of the CA is retained by the kidneys for elimination, but a certain amount, after a while, returns by the blood flow itself and is infused into the cardiac tissue again by the coronary arteries, i.e., the intravascular domain.

A reaction–diffusion–advection equation in a one-dimensional domain (of size *L*) is used to represent this behavior of the CA. The total amount of CA in the intravascular domain of the myocardium is imposed as an inflow into the 1D domain. The parameters of the equation were defined so that the time and the amount of flux at the output of the 1D domain represent the physiological behavior, such as the amount of CA retained by the kidneys and the time it takes to recirculate. Therefore, the amount of flow at the exit of the 1D domain is imposed as a recirculation parameter X(x,t) for the boundary condition of the CA flow in the intravascular domain.

The recirculation is therefore described by the following equation:(13)$$\frac{\partial C_{out}}{\partial t}+\nabla \cdot v_{out}C_{out}-\nabla \cdot \left(D_{out}\nabla C_{out}\right)+kC_{out}=0,\;\mathrm{in}\;\left[0,L\right],$$
where $v_{out}$ and $D_{out}$ represent the velocity (or convection term) and the diffusion, respectively, of this recirculatory system [7]. This equation is subject to the following conditions:(14)$$C\left(0,t\right)=\frac{\int_{\Omega_{i}}C_{i}d\Omega_{i}}{|\Omega_{i}|}.$$

The 1D recirculation model is subjected to appropriate boundary conditions for coupling with the intravascular model. The value C(0,t) in Equation (Equation 14) depicts the average CA concentration in the intravascular domain. It is used as a Dirichlet boundary condition at the left boundary of the 1D domain, and represents the coupling with the CA that leaves the myocardium. The coupling with the CA entering the myocardium (through the intravascular domain) is depicted by C(L,t), which corresponds to the CA obtained at the right boundary after the transport. This quantity is considered as a recirculation term X(x,t)=C(L,t), which is used for the boundary condition described in the following section. In other words, X(x,t) portrays the rise of CA concentration in the subsequent passages through the myocardium. Figure 1 illustrates the connection between the recirculation domain and the intra- and extravascular domains. It is important to highlight that the CA recirculation is added only in the root segments of arterial trees in the discrete model.

Physiologically, a portion of the CA is eliminated in the recirculation, especially by the kidneys. The model depicts it at the decay term $kC_{out}$. This phenomenon explains the reduction of the CA at the sequential passages in the myocardium.

### 2.7. Initial and Boundary Conditions

For the intravascular CA dynamics, a convective and diffusive flow of CA inflow into the intravascular domain through the epicardium is imposed and controlled by a transient Gaussian function, which is given by:(15)$$vC_{i}-D_{i}\nabla C_{i}=vQ\left(t\right),\;\mathrm{on}\;\Gamma_{epi},$$
with the function Q(t) defined as
(16)$$Q\left(t\right)=\frac{1}{\sigma \sqrt{2\pi }}e^{-\frac{1}{2}\left(\frac{t-t_{peak}}{\sigma }\right)}+X\left(t,\vec{x}\right),$$
where $\sigma^{2}$ reflects the variance of the CA infusion; $t_{peak}$ is the Gaussian mean, which is the peak value of the function; and $X(t,\vec{x})$ is the additional term generated by CA recirculation due to the cyclic behavior of the blood system.

Finally, no-flux boundary conditions are used for the other boundaries of the domain, as follows:(17)$$\begin{array}{cccc} D_{i}\nabla C_{i}\cdot n & =0\; & & \mathrm{on}\;\Gamma_{endo}, \end{array}$$(18)$$\begin{array}{cccc} D_{e}\nabla C_{e}\cdot n & =0\; & & \mathrm{on}\;\Gamma_{endo}, \end{array}$$(19)$$\begin{array}{cccc} D_{e}\nabla C_{e}\cdot n & =0\; & & \mathrm{on}\;\Gamma_{epi}. \end{array}$$

## 3. Numerical Methods

In this section, the numerical methods used for the discretization of the continuous porous-media model and for the discrete arterial tree model are presented.

### 3.1. Discretization of the Continuous Model

The discretization of the continuous domains (extravascular and fibrosis) of the hybrid model occurs in the same way as described in [7] for the continuous model. In summary, the finite volume method (FVM) was used to solve the Darcy equation as well as the advection–diffusion–reaction equations, where the advective term is handled via the third-order polynomial upwind scheme (TOPUS) to ensure stability and accuracy [28]. The explicit Euler method was used in the temporal part of the advection–diffusion–reaction equations.

### 3.2. Discretization of the Discrete Coronary Arterial Tree Model

It remains, therefore, to explain the methods used to apply the arterial tree provided by the CCO into the transport equation. It is noteworthy that the CCO method guarantees the conservation of mass in arterial trees.

The arterial tree model is a geometric substrate through which the CA will flow. It penetrates the tree through the root segment and divides at bifurcations until it reaches the terminal segments, where the CA flows into the tissue. The critical point of this step is to treat each segment of the arterial tree as one-dimensional, which means that the arterial tree is a series of one-dimensional interconnected segments, each one with flow and radius provided by the CCO. Therefore, differently from the continuous model presented in [7], in which the CA dynamics is treated as a two-dimensional flow, in the discrete part of the hybrid model, Equation (Equation 7) (reaction–diffusion–advection) is used in a one-dimensional network of segments.

The discretization of the skeleton provided by the CCO method was handled using the data structure of a graph. For such, we defined a constant parameter Δx, which represents the length of the discretization of each segment *j* of the tree. Figure 3 illustrates a simple graph with five nodes: a root, one regular, one bifurcation, and two terminals. In this case, initially, the structure has a list of nodes, and each one has a list of edges, i.e., each node has the information of the nodes it is connected to. The nodes of this structure represent the points of the discretized tree, and the edges represent the connections between the points. In the example presented in Figure 3, node 0 represents the root. It is connected to the node 1, which in turn is connected to the node 2, a bifurcation type.

The boundary condition of the arterial tree, given by Equation (Equation 16), is applied to the node 0. Through Equation (Equation 7), the CA is transported to node 1, followed by node 2. In this bifurcation point, the graph stores the list of edges 3, 4, and 1, i.e., node 2 has connections with outgoing nodes 3 and 4, and inlet node 1. The CA that leaves the bifurcation spread in terminal nodes 3 and 4. In this type of point, the CA leaks into the tissue. Using this system to store the variable of interest, each node has the information of which ones are its neighbors, capturing correctly from where the CA is coming and where it is going. By the end, given this communication system between the nodes of the discretized tree, we apply the FVM to Equation (Equation 7) to evaluate the CA propagation at the arterial tree. The details of this part can be found in [7]. However, the discretized equation will be different as the type of node.

For the regular node, 1, we have
(20)$$C_{1}^{t+1}=\left[D\frac{\left(C_{2}^{t}-2C_{1}^{t}+C_{0}^{t}\right)}{\Delta x^{2}}-v_{b}\frac{\left(C_{1}^{t}-C_{0}^{t}\right)}{\Delta x}\right]\Delta t+C_{1}^{t},$$
where $C_{i}^{t}$ is the concentration in node *i* at iteration *t*, $v_{b}$ is the velocity in face *b* and *D* is the diffusion coefficient (see Figure 4). Notice that we could use $v_{a}$ instead of $v_{b}$, since the velocity in a given segment *j* is constant.

For the node 2, which is a bifurcation node-type (see Figure 5), we have the equation
(21)$$\begin{array}{ccc} C_{2}^{t+1} & = & \left\{D\left[C_{1}^{t}-\left(1+\frac{r_{2}^{2}}{r_{1}^{2}}+\frac{r_{3}^{2}}{r_{1}^{2}}\right)C_{2}^{t}+\frac{r_{2}^{2}}{r_{1}^{2}}C_{3}^{t}+\frac{r_{3}^{2}}{r_{1}^{2}}C_{4}^{t}\right]\frac{1}{\Delta x^{2}}\right.\\ & - & \left.\left[\frac{C_{2}^{t}}{r_{1}^{2}}\left(v_{a}r_{3}^{2}+v_{b}r_{4}^{2}\right)-v_{c}C_{1}^{t}\right]\frac{1}{\Delta x}\right\}\Delta t+C_{2}^{t}. \end{array}$$
where $C_{i}^{t}$ is the concentration in node *i* at iteration *t*, while $r_{i}$ is the radius of the node *i*, and *D* is the diffusion coefficient. In this case, since the node 2 is a bifurcation, it has three neighbors, and the control volume used at the FVM is applied in three faces, *a*, *b*, and *c*, with the respective velocities $v_{a}$, $v_{b}$ and $v_{c}$.

For terminal nodes, such as node 3 in Figure 6, the following equation is considered:(22)$$C_{3}^{t+1}=\left[D\frac{\left(C_{2}^{t}-C_{3}^{t}\right)}{\Delta x^{2}}-v_{a}\frac{\left(C_{3}^{t}-C_{2}^{t}\right)}{\Delta x}-P\left(C_{3}^{t}-C_{e}^{t}\right)\right]\Delta t+C_{3}^{t},$$
where $C_{i}^{t}$ is the concentration in node *i* at iteration *t*, $v_{a}$ is the velocity at face *a*, and *D* is the diffusion coefficient. The variable $C_{e}^{t}$ represents the concentration on the extravascular medium at the closest point of the terminal node. This point is where the exchange of CA from intra- to extravascular media happens.

## 4. Numerical Experiments and Results

In this section, we present the setup of the numerical experiments and the simulations results conducted with the hybrid model for the three scenarios described before.

### 4.1. Setup of the Numerical Experiments

The heart has three main coronary arteries: the left circumflex artery (LCX), the left anterior descending (LAD), and the right coronary artery (RCA) [29]. Thus, we choose to generate a vascular system of three arterial trees such that each subtree has its root segment representing one of the coronary arteries mentioned above. The perfusion regions of each subtree possess the same area, since we considered the same perfusion flux for all of them. In the tree generation process was adopted the perfusion inflow $Q_{per}=1$ mL/min (see Figure 7) and $N_{term}$ = 355. The remain parameters are given by [17]: $p_{perf}$ = 100 mmHg; $p_{term}$ = 60 mmHg; γ = 3.

Analogously to what can be found in our previous work [7], the present work studies three scenarios: (1) healthy, (2) ischemia, and (3) infarction. However, while the first used the pressure boundary condition to simulate stenosis, here we changed the flow in a subtree of the domain. To simulate normal perfusion, ischemia, and myocardial infarction, we used different flow conditions at the arterial tree model, which are shown in panels (a), (b), and (c) of Figure 8 for the normal, ischemic, and infarction cases, respectively. In other words, the stenosis is depicted in the model through a branch with a reduced flow. Therefore, we chose a point at the tree to decrease its flow, to evaluate its impact on the nearby tissue. The point where we positioned the obstruction is indicated in Figure 8b,c. The magnitude of the stenosis is described by α, i.e., if α=2, the flow of the subtree is reduced by half; α=5 means the flow is divided by a factor of 5, and so on. It is important to highlight that the perfusion flow $Q_{perf}$ is kept constant at all three roots of the tree. This is also the same for the three scenarios explored here.

More details on the discretization of the continuum parts of the model (tissue and fibrosis) can be seen in [7]. We considered Δx= 0.25 mm for the arterial domain. Moreover, since we used the explicit Euler method for the discrete evaluation of the temporal derivative, it was necessary to satisfy the CFL condition.

For the healthy myocardium (Scenario 1, a value of α=1 was used. For the Scenario 2 (ischemia), a value of α=25 was applied, whereas for the Scenario 3 (infarction), α=30 was used. This last scenario also takes into account the presence of fibrosis at the damage region indicated in Figure 8d. For this, the other hypotheses described for Scenario 3 of the continuum model [7] are also applied here, i.e., used λ=1.0 was used, whereas λ= 0.25 was used for both healthy and ischemic scenarios. The same happens for $\lambda_{f}$ (fraction of fibrosis at the extravascular media): zero for healthy and ischemia scenarios; 0.5 for infarction. The porosity ϕ is also held in 0.10.

Aiming to adjust the qualitative results (MRI images) of the FP and LE perfusion exams, the remaining parameters were slightly changed when compared to those of the continuum model. Table 1 presents the values used in the simulations of this work.

### 4.2. Grid Independence Test

Before presenting the numerical results of fluid flow in the three cases of interest, we carried out a grid independence test to evaluate the stability and convergence of the numerical scheme adopted in this work. To this end, we considered the healthy scenario and evaluated it under three different mesh discretizations consisting of 100×100, 200×200, and 400×400 nodes. Simulations were carried out, and to evaluate the convergence of the numerical scheme, we computed the mean extravascular concentration over time for each case.

Figure 9 shows the results of the grid independence test, where one can observe that for the three meshes evaluated, the results have a high degree of convergence in terms of the response measured for the mean extravascular concentration.

### 4.3. Profiles of the CCO Arterial Trees

Figure 10 shows the profiles of flow, length, and radius of the segments of the arterial tree obtained by the CCO method.

### 4.4. Simulation Results

Next, we have the results of the hybrid model in Figure 11, which shows the three studied scenarios: healthy, ischemic, and infarcted, after 50 s (FP) and after 600 s (LE) of perfusion. In the healthy scenario, since there is no stenosis, perfusion occurs homogeneously in the whole tree. In the ischemic case, the model captures the phenomenon, since in the FP, the region of the obstruction (see Figure 8) is less perfused, and in the LE, the CA already moved out of the myocardium. For the third scenario, which considers the case of infarction, it is possible to observe the low perfusion at the FP exam, as well as CA confined at the obstructed region (as in fact, it happens during an MRI perfusion exam).

Figure 12 shows the evolution of the contrast agent concentration for the ischemia and for the infarction cases (check the Appendix A for the video of the entire simulation of the infarction scenario). One can clearly observe that in the infarction case (a), the CA remains confined in the obstructed region after the stenonis (indicated in Figure 8), whereas in the ischemia case (b), the CA flows out of that region.

## 5. Discussion, Conclusions and Future Works

Using CCO to construct an intravascular media based on arterial trees enables essential investigations. For instance, it allows one to choose the point of the tree where the flow will be purposefully reduced to represent stenosis. In other words, it is viable to study the impact of the reduced perfusion flow in different branches of the tree. In particular, in this work, we chose a branch of the arterial tree close to the right side of the endocardium to reduce the flow for the scenarios of ischemia and infarction, which had its flow reduced by a factor of 25 and 30, respectively. In each case, the results generated by the hybrid model (see Figure 11) are compatible with the respective clinical image.

In the scenarios presented in this paper, we opted to construct the vascular system using three trees, each one aiming to represent the branches arising from the main coronary arteries (LCX, LAD, and RCA). The CCO method can be used to generate more trees for the system, since each subdomain of perfusion of each tree has been defined. This allows for the study of different configurations of arterial trees.

In addition, from the perspective of the CCO method possibilities, we applied the same influx (1 mL/min) for each tree (LCX, LAD, and RCA). In the next section, we discuss the future perspective for applying the CCO varying these two principal features: the subdomains of perfusion and the influxes.

The arterial trees generated by the CCO method, which is based on optimization principles, closely resemble real arterial trees, both visually and also with respect to morphometric parameters. In this paper, we used the Hagen–Poiseuille equation to describe pressure and flux in the arterial tree. Therefore, the only geometrical feature that affects the simulations is the radius of each segment of the arterial tree. Nevertheless, our results show that by prescribing the radius and/or flux along the tree, we can simulate different degrees of arterial stenosis. The presented framework can be used to integrate data obtained by computerized fractional flow reserve (cFFR). Traditionally, in cFFR, the complex geometry of the coronary tree and the Navier–Stokes equations are used to compute the fluid dynamics [31]. The resulting fluxes computed via cFFR can be used as input in our hybrid model.

Concerning fluid flow modeling, some details are worth discussing. It is important to highlight that the interstitial domain exhibits fluid flow. However, we neglected the convection term in Equation (Equation 9) in this work. To model this phenomenon adequately would necessitate the inclusion of an additional equation for interstitial pressure, accounting for time-varying cavity pressures and plasma flow into the lymphatic system, as presented in previous works [32,33]. Conversely, at a macroscopic level, the vascular tree densely covers the tissue, facilitating oxygen diffusion—or in our case, the contrast agent (CA)—to effectively reach all regions of the interstitial domain. Hence, to streamline our model, we have opted not to include the interstitial domain’s pressure dynamics explicitly. Instead, our primary focus is on modeling the diffusion-driven movement of the CA in the extravascular domain. Most importantly, this choice was not arbitrary, but rather driven by the availability of clinical data. Specifically, we have access to crucial clinical data such as pressure and flux measurements within the intravascular domain obtained through fractional flow reserve assessments and information on contrast dynamics within both the intra- and extravascular domains. The clinical data support our choice of not explicitly incorporating the interstitial pressure dynamics.

In this context, it is also important to recall the Kedem–Katchalsky equations [34], which describe the fluxes of a solution ($J_{v}$) and of a solute ($J_{s}$) across a membrane. The flux $J_{s}$ depends on the concentration and pressure differences across the membrane. In this work, we simplify our model by not taking into account the interstitial pressure. Therefore, as a simplification, $J_{s}$ only depends on the concentration difference.

### 5.1. Validation of the Results

In summary, the new hybrid model qualitatively reproduces the CA dynamics and patterns found in patients with both ischemia and infarction in the same way as shown in our previous work [7]. In addition, by modeling the intravascular domain as a discrete arterial model, it establishes a way to study how particular and local features of the flow in the coronaries of a patient (stenosis or stent) influence the dynamics and patterns of cardiac tissue perfusion.

Nevertheless, we acknowledge the importance of a final quantitative validation, which could be possible using clinical data derived from the fractional flow reserve and cardiac perfusion MRI exams. This integration allows for the incorporation of cFFR data derived from CT scans with the discrete arterial tree component of the model, as well as data from cardiac perfusion MRI exams into the continuum part of the model. We aim to incorporate such data in the future to further validate and strengthen our method.

Integrating cFFR measurements facilitates the assessment of the significance of coronary narrowings and their impact on blood flow to the heart. When combined with cardiac perfusion MRI data, a comprehensive view of coronary circulation, myocardial perfusion, and tissue characteristics can be achieved.

This integration will offer two key benefits. Firstly, it enables a more accurate and detailed evaluation of the patient’s cardiac status, providing clinicians with the necessary information to make informed decisions regarding interventions such as angioplasty or stenting. Secondly, it enhances the understanding of cardiovascular disease by offering a comprehensive assessment of cardiac function, anatomy, and perfusion.

### 5.2. Limitations and Future Works

It is worth mentioning that one of the main limitations of our model is the two-dimensional flow. We used 1 mL/min for the inflow at the three coronary arteries here simulated. However, even if the transmural direction has the higher magnitude of perfusion, i.e., from epicardium to endocardium, we still have a considerable amount of fluid that perfuses from the apex to the base of the heart [35]. Therefore, in future works, we aim to extend the hybrid model for three dimensions. Thus it will be possible to capture, in a more reliable sense, the anisotropy of myocardial arteries, thus giving one more step in order to provide an appropriate model for the use of physicians. Still, from the perspective of model improvements, we intend to use the CCO method to also construct a model of the venous arterial tree. Within this context, it will be required to prescribe outfluxes of the myocardium instead of the decay parameters $k_{e}$ and $k_{f}$ of the Equations (Equation 9) and (Equation 11), respectively. The outfluxes are parameters that can be found in the literature [36], and it will decrease the overfitting of our model.

Regarding the CCO method, it ensures that a fixed value of $Q_{perf}$ applied in different arterial trees implies that the perfused regions for each tree will have similar areas [21]. Therefore, in the scenarios where we studied the CA dynamics, the three regions of perfusion were previously defined with the same area, as described in Figure 7. Thus, when we prescribe the perfusion flow $Q_{perf}=$1 mL/min for the three myocardial trees, the condition mentioned above is guaranteed. In future works, we intend to refine the method currently used, adapting the implementation in a way that, once having defined the perfusion flow, the regions of perfusion will be automatically adjusted. When doing so, it will be possible, for instance, to properly regulate the blood volume that travels through each coronary artery [37].

In the end, the new model proposed here has some advantages to quantifying cardiac perfusion. However, due to the high number of parameters and equations, and for the purpose of investigating how they affect the response of cardiac perfusion of specific patients, we planned to elaborate a clinical study similar to those presented in [38,39,40]. In other words, given clinical images of ischemia and infarction provided by MRI exams, we intend to apply the model presented here to indicate possible obstructions of the intravascular media that would be causing the perfusion problem.

### 5.3. Conclusions

The new hybrid model enables simulations that establish a connection between blood flow, stenosis in the coronary arterial tree, contrast agent dynamics, and myocardial tissue perfusion. This integration allows for the incorporation of cFFR data derived from CT scans with the discrete arterial tree component of the model, as well as data from cardiac perfusion MRI exams into the continuum part of the model. The primary objective of this integration is to elucidate the relationship between the spatial–temporal dynamics of the contrast agent in the myocardium and the blood flow in the coronary arterial tree, which has significant clinical implications.

Furthermore, integrating cFFR measurements facilitates the assessment of the significance of coronary narrowings and their impact on blood flow to the heart. When combined with cardiac perfusion MRI data, a comprehensive understanding of coronary circulation, myocardial perfusion, and tissue characteristics can be achieved. The integration of these data sources offers two key benefits. Firstly, it enables a more accurate and detailed evaluation of the patient’s cardiac status, providing clinicians with essential information to make informed decisions regarding interventions such as angioplasty or stenting. Secondly, it enhances the understanding of cardiovascular disease by offering a comprehensive assessment of cardiac function, anatomy, and perfusion.

In summary, the proposed hybrid model, together with the integration of cFFR data and cardiac perfusion MRI data, holds great promise for advancing our understanding of cardiac perfusion dynamics and improving clinical decision making in the management of cardiovascular disease. By establishing a comprehensive link between blood flow, stenosis, contrast agent dynamics, and myocardial tissue perfusion, this integrated approach has the potential to significantly impact patient care and contribute to enhanced outcomes in cardiovascular medicine.

## Figures and Tables

**Figure 1 entropy-25-01229-f001:**
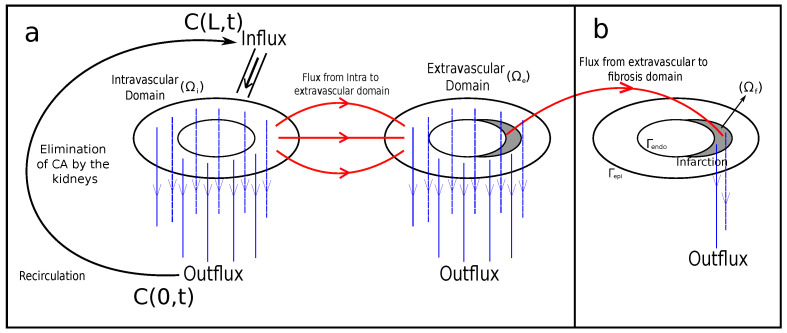
Model of two domains (**a**), used to simulate the scenarios of (1) healthy and (2) ischemic myocardium; and the model with three domains (**b**). The third domain is used to simulate a scenario of (3) infarction.

**Figure 2 entropy-25-01229-f002:**
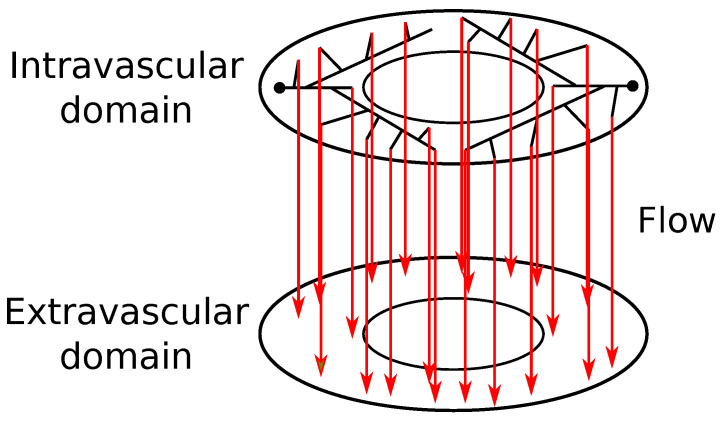
Communication between intravascular domain, built using CCO method, and the extravascular one; continuum. This communication takes place at the terminal segments of the arterial tree.

**Figure 3 entropy-25-01229-f003:**
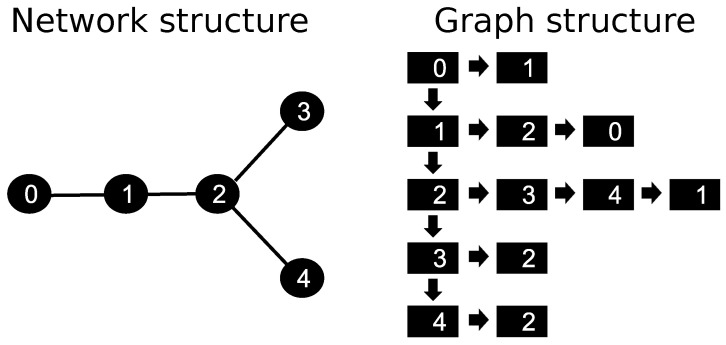
Data structure used to represent the arterial tree provided by the CCO method. On the left, the figure represents the network of points obtained from the spatial discretization of size Δx, made upon the arterial tree. The figure on the right shows the graph structure: there is a linked list of nodes, and each node has a list of edges. This way, each node has the information on where the CA is coming from and to which neighbors the flow is going to be directed.

**Figure 4 entropy-25-01229-f004:**
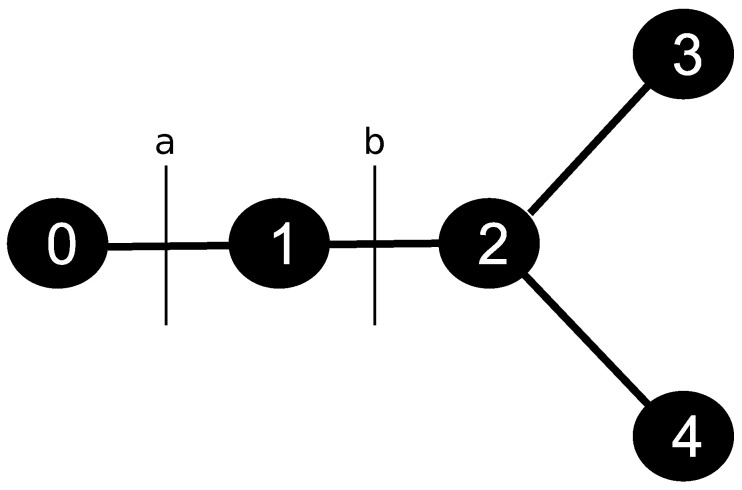
Node 1, of the regular type, i.e., has only two neighbours, 0 and 2, and the respective faces used at the FVM, *a* and *b*.

**Figure 5 entropy-25-01229-f005:**
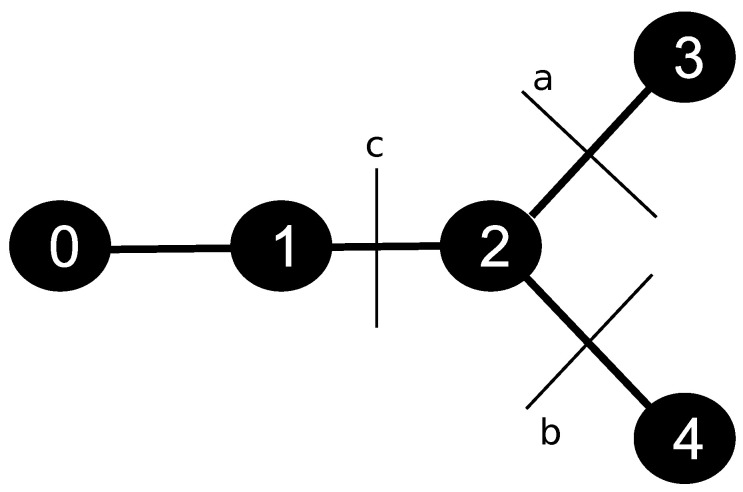
Node 2, of the type bifurcation, i.e., it has three neighbours, 1, 3 and 4, and the respective faces used at the FVM, *a*, *b* and *c*.

**Figure 6 entropy-25-01229-f006:**
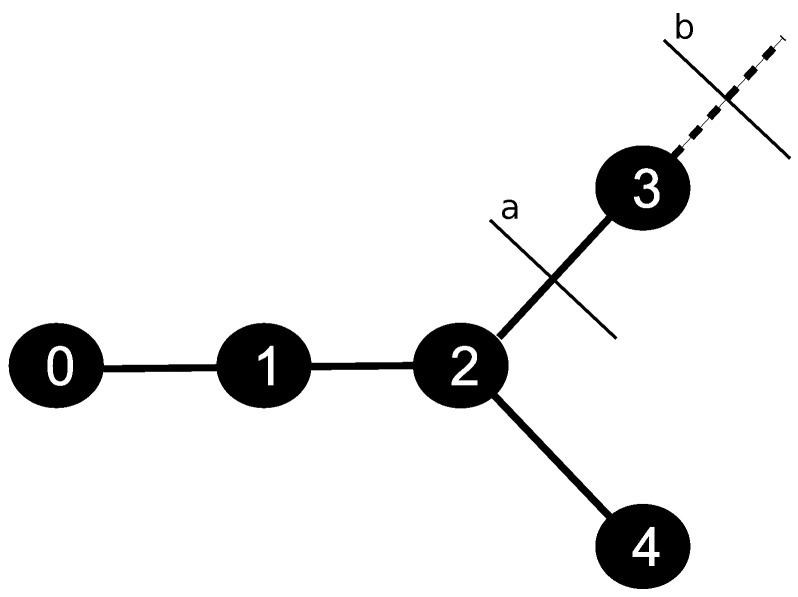
Node 3, of the terminal type, i.e., it has only one neighbour, node 2, and the respective faces *a* and *b* used at the FVM scheme.

**Figure 7 entropy-25-01229-f007:**
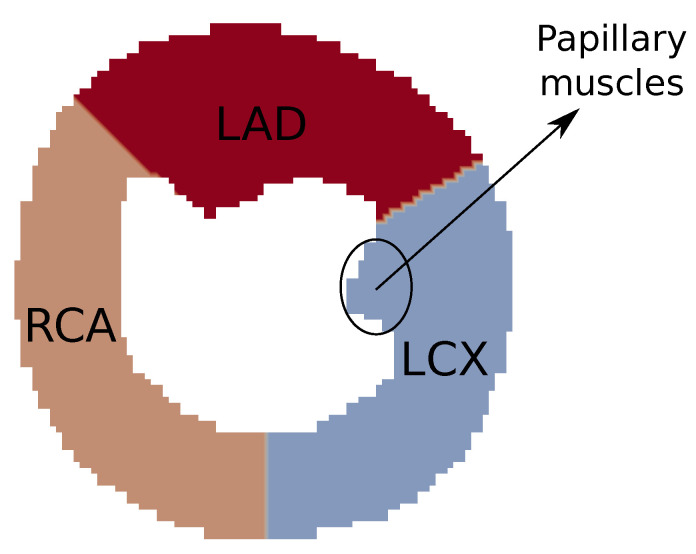
Simplified representation of the regions perfused by the trees from coronary arteries: LCX, LAD, and RCA. It is also indicated the position of the myocardial papilary muscles. Adapted from [30].

**Figure 8 entropy-25-01229-f008:**
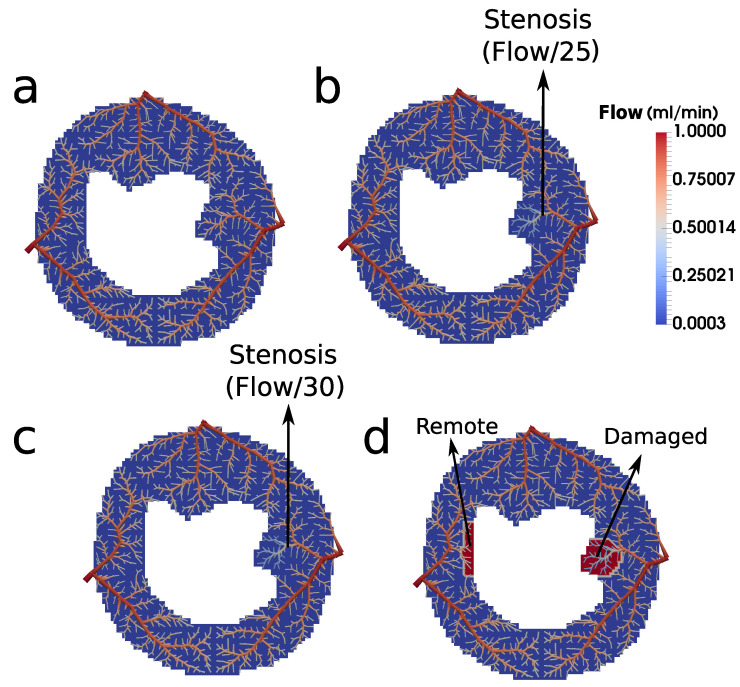
Results of the simulations carried out: (**a**) healthy, (**b**) ischemia, and (**c**) infarction. (**b**,**c**) indicate the stenosis position, where a reduced flow is imposed. For the ischemic case, the flow was decreased by a factor of 25 (α=25), whereas for the infarction, it was decreased by a factor of 30 (α=30). Panel (**d**) shows the regions of interest (remote and damaged) for the evaluation of the signal intensity of the contrast agent.

**Figure 9 entropy-25-01229-f009:**
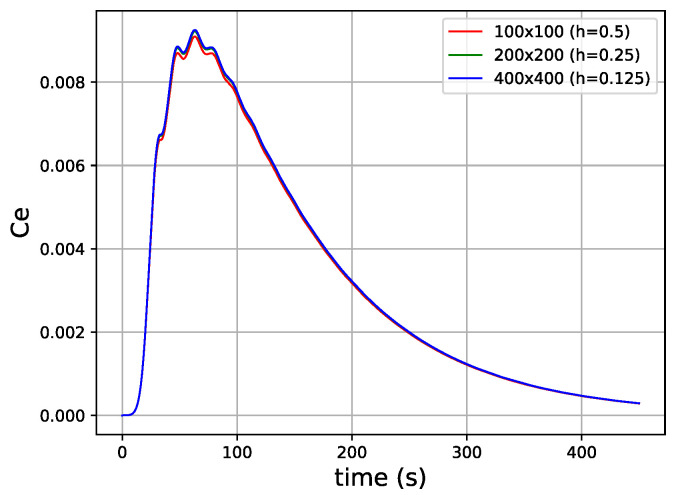
Results of the grid independence test in terms of the total extravascular concentration as a function of time for meshes consisting of 100×100, 200×200, and 400×400 nodes.

**Figure 10 entropy-25-01229-f010:**
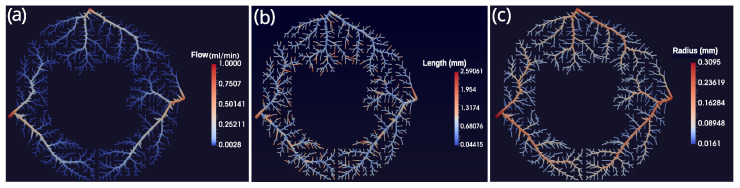
Profiles of (**a**) blood flow (mL/min); (**b**) length of the segments; and (**c**) radius of the segments for the arterial trees provided by the CCO method.

**Figure 11 entropy-25-01229-f011:**
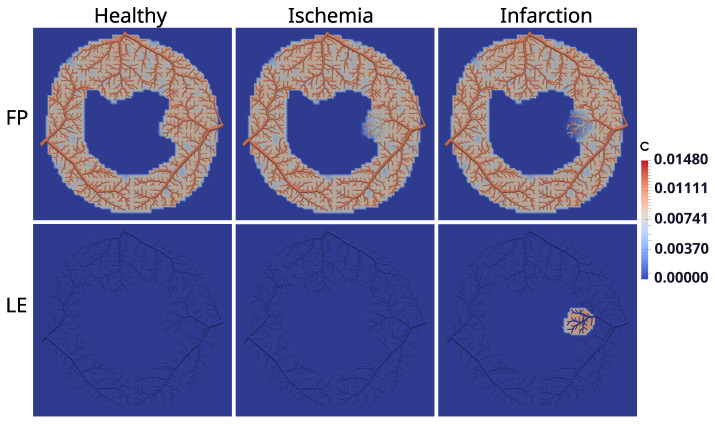
Dynamics of the CA at the end (50 s) of the exams first pass (FP) and after the late enhancement (LE) at 600 s.

**Figure 12 entropy-25-01229-f012:**
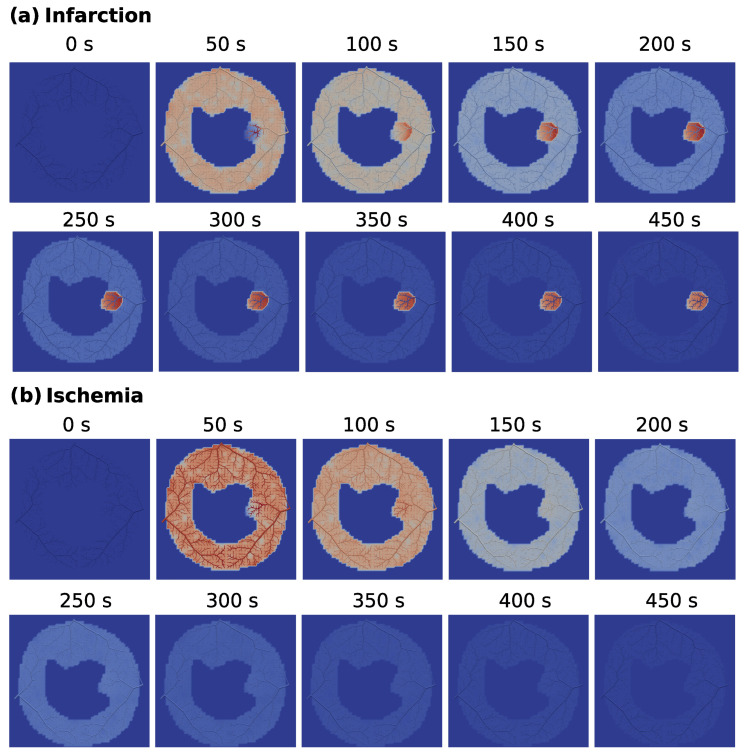
Contrast agent concentration for the (**a**) infarction and (**b**) ischemia scenarios at every 50 s until the final time of 450 s.

**Table 1 entropy-25-01229-t001:** Parameters used in each scenario of the model.

Parameter (Units)	Description	Healthy/Ischemic	Infarction
$P\;(s^{-1})$	endothelial permeability	1.0	1.0
$k_{e}$ ($s^{-1}$)	decay parameter	0.007	0.002
$k_{f}\;\left(s^{-1}\right)$	decay parameter (fibrotic)	0	0.001
$k_{ef}\;\left(s^{-1}\right)$	rate that contrast moves from the interstitium to the fibrosis	0	0.01
ϕ (−)	porosity	0.10	0.10
λ (−)	fraction of the extravascular domain that is occupied by the interstitial space	0.25	1.0
$\lambda_{f}\;\left(-\right)$	fraction of fibrosis at the extravascular media	0	0.5
$D\;({\mathrm{mm}}^{2}s^{-1})$	diffusion coefficient	$10^{-2}$	$10^{-2}$
σ	variance of the CA infusion	6.0	6.0
$t_{peak}\;\left(s\right)$	Gaussian mean	25	25
$v_{out}\;\left(\mathrm{mm}\cdot s^{-1}\right)$	velocity of the recirculatory system	0.06	0.06
$D_{out}\;\left({\mathrm{mm}}^{2}s^{-1}\right)$	diffusion coefficient of the recirculatory system	0.05	0.05
$k\;(s^{-1})$	contrast agent clearing rate	0.01	0.01

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
