# Peer review of "A Hybrid Model for Cardiac Perfusion: Coupling a Discrete Coronary Arterial Tree Model with a Continuous Porous-Media Flow Model of the Myocardium"

_entropy, 2023, doi:10.3390/e25081229_

Round 1

Reviewer 1 Report

This paper proposes a hybrid approach combining a detailed model of the coronary arterial tree with a porous-media flow model of the myocardium to study cardiac perfusion. The model can simulate blood flow and the dynamics of a contrast agent used in MRI exams. It can also simulate different scenarios, including normal perfusion and ischemia, and has the potential to improve the understanding and treatment of cardiovascular disease. However, besides some undefined term in the context, there are some doubts about the mathematical model needed to be cleared before my recommendation of publishing this paper.

(1)   No definitions of FP and LE, when they first show up in the context.

(2)   Is the porous medium described in Eq. (6) for extravascular region or intravascular region? From Eq. (9), it seems the porous medium indicating intravascular region, since it uses velocity v described in Eqs. (7) and (8) for convection.

(3)   What is the source term alpha in Eq. (8)? Where does it come from? There is no further description of it after Eq. (8).

(4)   The f term in Eq. (9) describes the extravasation of CA. From Eq. (11), it only depends on the concentration difference in intra- and extra-vascular regions. From Kedem-Katchalsky equation, there should be an additional term about the extravasation leakage of CA related to Starling equation.

(5)   Why there is no convection term in Eq. (10)? There should be fluid flow in interstitial region.

Reviewer 2 Report

This paper presents a model that describes the perfusion of contrast agents in cardiac tissue by coupling a discrete coronary arterial network model with a porous-media flow model. The paper is well-written, but a validation of the model is missing.

·      Vessel compliance is not considered in the model. Could this affect the results?

·      The fluid dynamics in the coronary tree also depends on the geometry. How do you take it into account?

·      The validation of the results obtained is missing.

·      In table 1 it would be advisable to insert the description of the parameters.

Reviewer 3 Report

What is the main objective of the paper?

What does the proposed hybrid approach for cardiac perfusion entail?

Add in introduction

The papers selected for review in this study represent diverse areas of biomedical research. Zeng et al. (2020) propose a novel method to enhance the sensitivity of hyperpolarized xenon nuclear magnetic resonance (NMR) imaging using metal-organic framework entrapment. Zhang et al. (2022) investigate the role of neurogenesis and neural stem/progenitor cell proliferation in stroke models, highlighting the potential of the FOXO3a/p27Kip1 axis as a therapeutic target. Hao et al. (2022) introduce a new approach for evaluating cardiac ischemia/reperfusion using serum metal ion-induced cross-linking of photoelectrochemical peptides and proteins. Zhou et al. (2022) present enzyme-free and enzyme-resistant detection methods for complement component 5, offering a valuable diagnostic tool for acute myocardial infarction. Xue et al. (2022) explore the impact of cardiomyocyte-specific knockout of ADAM17 on diabetic cardiomyopathy, revealing potential therapeutic strategies. Tian et al. (2022) investigate the contribution of gut microbiome dysbiosis to abdominal aortic aneurysm through neutrophil extracellular trap formation. These studies collectively contribute to advancements in diagnostic techniques, treatment strategies, and understanding the underlying mechanisms of various medical conditions. doi: 10.1073/pnas.2004121117; doi: 10.1007/s12035-021-02710-5; doi: 10.1021/acssensors.1c02305; https://doi.org/10.1016/j.snb.2022.132315; doi: 10.1038/s41392-022-01054-3; doi: https://doi.org/10.1016/j.chom.2022.09.004

How is the coronary arterial tree network modeled, and what is the significance of the constructive constrained optimization (CCO) algorithm?

Which mathematical model is used to describe blood flow in the coronary arterial network?

How is the flow of a contrast agent in the myocardium modeled, and what equations are utilized?

What specific aspect of the contrast agent's interaction with fibrosis is considered in the model?

How is the tissue domain represented in the model, and what type of media is used to describe it?

What additional feature is incorporated into the model regarding the recirculation of the contrast agent?

What types of numerical experiments were conducted to test the proposed approach?

How does the model simulate normal perfusion, endocardial ischemia, and myocardial infarction?

What is the potential clinical application of the proposed model?

How does the proposed study aim to integrate information from different exams, and what benefits can it provide in terms of understanding and treating cardiovascular disease?

Moderate editing of English language required

Reviewer 4 Report

This paper presents an application of the classical of the hydrodynamic equations, such as continuity, Navier-Stokes, and convective-diffusion, to the problem of the blood flow in the vascular. I think that that this paper can be published after minor revision. Since the main results of this paper are based on the numerical modelling, their accuracy should be proved

1) What numerical method has been applied to solve a system of equation? Or it was any commercial software?

2) Please, provide numerical validation of the numerical algorithm.

3) Please, provide a test of the effect of grid refinement on the stability of numerical results.

Author Response

Find attached the response letter.

Round 2

Reviewer 2 Report

A validation of the model is missing. The considerations made by the authors are not sufficient.

Reviewer 3 Report

Comments is not explain in according to my comments 

Add in introduction

The papers selected for review in this study represent diverse areas of biomedical research. Zeng et al. (2020) propose a novel method to enhance the sensitivity of hyperpolarized xenon nuclear magnetic resonance (NMR) imaging using metal-organic framework entrapment. Zhang et al. (2022) investigate the role of neurogenesis and neural stem/progenitor cell proliferation in stroke models, highlighting the potential of the FOXO3a/p27Kip1 axis as a therapeutic target. Hao et al. (2022) introduce a new approach for evaluating cardiac ischemia/reperfusion using serum metal ion-induced cross-linking of photoelectrochemical peptides and proteins. Zhou et al. (2022) present enzyme-free and enzyme-resistant detection methods for complement component 5, offering a valuable diagnostic tool for acute myocardial infarction. Xue et al. (2022) explore the impact of cardiomyocyte-specific knockout of ADAM17 on diabetic cardiomyopathy, revealing potential therapeutic strategies. Tian et al. (2022) investigate the contribution of gut microbiome dysbiosis to abdominal aortic aneurysm through neutrophil extracellular trap formation. These studies collectively contribute to advancements in diagnostic techniques, treatment strategies, and understanding the underlying mechanisms of various medical conditions.

Moderate editing of English language required

Author Response

We thank the reviewer for this valuable suggestion on the literature review. The references closely related to the present work were included in an appropriate literature review paragraph in the introduction, as suggested by the reviewer.  Please check the revised manuscript, section 1 (marked in red).

Reviewer 4 Report

-

Round 3

Reviewer 2 Report

The validation of the results, in my opinion, is inconsistent.

Reviewer 3 Report

Accept